# Bridging Gaps in Obesity Assessment: Spanish Validation of the Eating Behaviors Assessment for Obesity (EBA-O)

**DOI:** 10.3390/nu17142344

**Published:** 2025-07-17

**Authors:** María José Jaen-Moreno, Matteo Aloi, Ana Alcántara-Montesinos, Ana Jiménez-Peinado, Cristina Camacho-Rodríguez, Elvira Anna Carbone, Marianna Rania, Marcela M. Dapelo, Fernando Sarramea, Cristina Segura-Garcia, María José Moreno-Díaz

**Affiliations:** 1Maimonides Biomedical Research Institute of Cordoba (IMIBIC), 14004 Córdoba, Spain; 2Department of Morphological and Sociosanitary Science, University of Cordoba, 14004 Córdoba, Spain; 3Psychiatric Unit, Reina Sofia University Hospital, 14004 Córdoba, Spain; 4Department of Clinical and Experimental Medicine, University of Messina, 98122 Messina, Italy; matteo.aloi@unime.it; 5Department of Health Sciences, University “Magna Graecia” of Catanzaro, 88100 Catanzaro, Italy; elvira.carbone@unicz.it; 6Outpatient Unit for Clinical Research and Treatment of Eating Disorders, University Hospital Renato Dulbecco, 88100 Catanzaro, Italy; 7Mental Health Services, Obesity and Diabetes Program, Clinica Universidad de los Andes, Santiago 7620157, Chile; 8Department of Medical and Surgical Sciences, University “Magna Graecia” of Catanzaro, 88100 Catanzaro, Italy

**Keywords:** obesity, eating behavior, assessment, psychometric properties, test

## Abstract

**Background and Objective**: Obesity is currently one of the major challenges in medicine. Research indicates that assessing eating habits can contribute significantly to the development of more effective treatment. This study aims to validate the Eating Behaviors Assessment for Obesity (EBA-O) in a sample of Spanish adults with overweight or obesity. **Methods**: This cross-sectional study included 384 participants. To evaluate the structure, reliability, and measurement invariance of the Spanish EBA-O, we conducted a confirmatory factor analysis (CFA), calculated McDonald’s omega for reliability, and carried out a hierarchical sequence of multigroup CFAs. Two-way MANOVA was used to assess the effects of sex and body mass index (BMI) categories on EBA-O scores. **Results:** CFA supported a second-order five-factor structure for the EBA-O, demonstrating excellent fit indices. It respected the configural, metric, and scalar invariance. The Spanish version of the EBA-O exhibited significant correlations with measures of binge eating, food addiction, and eating disorder psychopathology. Internal consistency was high (ω = 0.80). Significant effects of sex and BMI were observed across EBA-O subscales. **Conclusions**: The EBA-O appears to be a valid, reliable, and easy-to-use instrument for assessing eating behaviors among Spanish-speaking individuals with overweight or obesity. Its strong psychometric properties support its use in both clinical settings and research, enhancing the development of tailored interventions for this population.

## 1. Introduction

Obesity is a complex metabolic disease with significant implications beyond aesthetics [1,2,3,4], and it represents a global health challenge affecting more than 2.5 billion adults worldwide. Effective treatment requires a comprehensive approach, incorporating interventions that range from nutritional education to surgical procedures [5,6].

Although not considered a mental disorder by the diagnostic classification criteria (DSM-5 [7] or ICD-11 [8]), obesity frequently co-occurs with psychiatric syndromes, resulting in substantial clinical, social, and economic burdens [9]. Among the complex and bi-directional associations between obesity and mental health, regardless of the nature of these associations, the most prominent is its link between obesity and binge eating disorder (BED) [10,11,12,13,14].

Consequently, a thorough assessment of the dysfunctional eating behaviors in individuals with obesity is critical for effective clinical management.

Obesity-related eating behaviors are heterogeneous and may lead to different outcomes [15,16,17,18,19], which has drawn increasing clinical and research attention. Behaviors such as binge eating [20], grazing [21,22,23], night eating [24,25], food addiction [26,27], sweet eating [28], and hyperphagia [29] are among the most observed in this population and contribute significantly to the complexity of disordered eating [30]. Notably, the study by Caroleo et al. [31] highlighted the clinical relevance of differentiating these behaviors, as individuals with obesity can be phenotyped based on their specific eating patterns.

However, assessing these varied eating behaviors can be challenging in clinical settings. Currently available tools often require the use of multiple scales, each designed to evaluate a single behavior and typically tailored to patients with eating disorders rather than those with obesity. This fragmented approach is time-consuming and may be difficult to interpret, especially for clinicians lacking specialized training in eating disorders. Therefore, there is a clear need for a comprehensive, reliable, and easy-to-use instrument to streamline the assessment of disordered eating behaviors in this population.

To address this need, Segura-García et al. [32] developed and validated the Eating Behavior Assessment for Obesity (EBA-O), an 18-item questionnaire designed to assess the presence and severity of five key eating behaviors common in obesity (i.e., night eating, food addiction, sweet eating, hyperphagia, and binge eating). The EBA-O has shown strong psychometric properties and is available in Italian [32] and Greek [33]. Both the Italian and Greek versions agree with the 18-item second-order five-factor model and have demonstrated that the EBA-O has good internal consistency (McDonald’s ω ranging from 0.80 to 0.92), good convergent validity with other measures of eating psychopathology, and good discriminant validity.

To the authors’ knowledge, no other Spanish instrument specifically designed and validated for people with BMI ≥ 25 that simultaneously assesses the five eating behaviors examined by the EBA-O (i.e., night eating, food addiction, sweet eating, hyperphagia, binge eating) exists. So, despite its potential utility, a validated Spanish version of the EBA-O is lacking. This study aims to fill this gap by examining the psychometric properties of the Spanish adaptation of the EBA-O in a sample of overweight and obese individuals from the Spanish speaking population.

## 2. Materials and Methods

### 2.1. Participants and Procedures

Participants were recruited among general population through targeted announcements on social media platforms (e.g., Facebook, Instagram, and X of the primary researchers, M.J.J.-M. and A.A.-M.) between February and May 2023. Data about the purpose of the study, voluntariness of participation, and management and retention of data were provided at the beginning of the survey. Specifically, participants were informed that no identifying data (e.g., name, location, IP) would have been requested, collected, or stored and that data would be used in aggregate. Eligibility screening was also embedded at the beginning of the survey. Participants were then asked to click on the consent box before further proceeding with the psychometric questionnaires. The convenience sample was required to report (a) age between 18 and 65; (b) self-identifying Spanish as the primary language; (c) self-reported body mass index (BMI) of 25 kg/m^2^ or higher; (d) absence of current diseases or treatments potentially affecting metabolic processes leading to weight gain or loss; (e) not being pregnant or breastfeeding over the last 12 months; (f) completion of all self-report questionnaires; and (g) valid informed consent. To this extent, the survey collected demographics (i.e., age, sex) and socio-demographic characteristics (i.e., education level, employment status). It also gathered information on general medical history (including current or past mental health disorders and active treatments), and participants were asked to self-report their current weight and height for body mass index calculation (BMI; weight (kg)/[height (m)]^2^). After completing the initial screening survey, participants proceeded completing the psychometric battery detailed in Section 2.2. A post hoc filtering procedure was applied to further test for eligibility, excluding data from individuals who did not meet the inclusion criteria.

This study received ethical approval from the Ethics Committee for Human Research of the University of Córdoba (reference CEIH 23–23) and was carried out following the latest revision of the Declaration of Helsinki.

### 2.2. Measures

The Eating Behaviors Assessment for Obesity (EBA-O), originally developed and validated by Segura-García et al. [32], underwent a double Italian/Spanish forward/backward translation process. Following an initial agreement among translators, a researcher blinded to the original version back translated the instrument into Italian to ensure accuracy. Neutral, standard Spanish was employed throughout the translation process and to enhance the instrument’s applicability across a broader range of Spanish-speaking populations, the preliminary translation was subsequently administered to by a group of 15 volunteers from various Spanish-speaking countries (i.e., Argentina, Chile, Colombia, Mexico, and Spain). All raters considered the items clear and easy to understand, with only minor adjustments made to enhance cultural sensitivity across different Spanish-speaking populations. Legal authorization for translation and adaptation was obtained from the original author.

The EBA-O comprises 18 items rated on an 8-point Likert scale (ranging from 0 = never to 7 = every day) and is designed to assess the presence and severity of five pathological eating behaviors commonly associated with obesity (i.e., night eating, food addiction, sweet eating, hyperphagia, and binge eating) over the preceding three months (Appendix A).

To assess the convergent validity of the Spanish version of the EBA-O, the following psychometric questionnaires with established validation studies conducted in Spanish-speaking populations were administered:The Binge Eating Scale (BES) [34], a 16-item self-report measure that assesses binge eating severity. Scores are interpreted as follows: <17 (unlikely), 17–27 (possible), and >27 (probable) risk of binge eating disorder. McDonald’s ω for the BES in this study was 0.88.The Yale Food Addiction Scale (YFAS) [35], used to assess addiction-like eating behavior over the past 12 months. This 35-item scale, scored on an 8-point Likert scale (0 = never to 7 = every day), measures 11 symptoms of food addiction. The YFAS 2.0 provides the following two scoring methods: (a) total number of criteria met (range: 0–11) and (b) severity level based on DSM-5 criteria for Substance Use Disorder (mild: 2–3 symptoms, moderate: 4–5 symptoms, severe: 6 or more symptoms). A “food addiction diagnosis” requires meeting at least two criteria plus experiencing impairment or distress. The Kuder–Richardson reliability coefficient for the YFAS 2.0 in this study was 0.87.The Eating Disorder Examination-Questionnaire (EDE-Q) [36], a 28-item self-report questionnaire that assesses eating disorder psychopathology over the past four weeks. It yields four subscale scores (restraint, eating concern, weight concern, and shape concern) and a total score. McDonald’s ω values for the EDE-Q subscales in this study were Restraint (0.76), Eating Concern (0.78), Weight Concern (0.77), and Shape Concern (0.84). The McDonald’s ω for the total score was 0.85.

### 2.3. Statistical Analysis

Initially, in line with recommendations for validation studies [37,38,39], a minimum sample size of 300 participants was pointed to allow statistical adequacy for the analyses. Additionally, according to MacCallum et al. [39] based on power analysis using RMSEA-based criteria, a sample size of 380 was deemed sufficient, assuming moderate-to-high factor loadings (≥0.5) and model degrees of freedom ≈ 130. The power to detect close versus poor fit (RMSEA 0.05 vs. 0.08) with α = 0.05 was estimated to be >0.85, indicating that the sample size was adequate to detect model misfit and ensure stable parameter estimates [40].

The present study applied CFA to examine the factor structure of the Spanish version of the EBA-O, in accordance with best practices for validating theoretical models. According to Brown (2006) [41] “unlike the EFA, the CFA requires a solid empirical or conceptual basis to guide the specification and evaluation of the factor model” (p. 12). Since the original EBA-O was developed based on a well-defined theoretical framework and has already demonstrated robust psychometric properties both in its original Italian version [32] and in its subsequent Greek adaptation [33], the use of CFA is justified. Furthermore, CFA is particularly suitable for testing the applicability of a previously validated model in new populations, including cross-cultural contexts, as long as the theoretical basis of the model remains relevant [42]. Thus, a second-order five-factor model was tested using confirmatory factor analysis (CFA) with the open-source JASP software (version 0.18.3) [43]. This analysis aimed to assess the underlying factor structure of the EBA-O and validate the suitability of a total score. The diagonally weighted least squares (DWLS) estimator, using a polychoric correlation matrix, was selected, as it is the most appropriate method for modeling ordered data.

The following indices were used to evaluate the model fit: the relative chi-square (χ^2^/*df*), Tucker–Lewis index (TLI), comparative fit index (CFI), root mean square error of approximation (RMSEA), and standardized root mean square residual (SRMR). Adequate values were defined as TLI and CFI ≥ 0.90 (adequate) and ≥0.95 (very good), RMSEA ≤ 0.08 (adequate) and ≤0.05 (very good), and SRMR close to 0.08. Additionally, a good fit was indicated by χ^2^/*df* values < 3.0 and a very good fit by values < 2.0, following the guidelines proposed by Hu and Bentler [44].

We also run a multiple-group CFA to establish the measurement invariance of the EBA-O across sex (i.e., men and women) and BMI (i.e., overweight and obese) using the software RStudio R 4.4.1 [45]. The process was carried out in three steps. First, a baseline configural invariance model was established to check if the factor structure was consistent across the groups. Then, the metric invariance model was established to ascertain if the factor loadings for the items were the uniform for sex and BMI. This is crucial for significant comparison between groups [46]. Lastly, the scalar invariance model was carried out to test if the factor loadings and intercepts were the same across the groups. Due to the nested nature of these three models, measurement invariance was determined based on the overall model fit and changes in fit indices between them. Measurement invariance is supported if the comparison between the two models meets a non-significant Δχ^2^, ΔRMSEA < 0.050, ΔCFI < 0.004, and ΔSRMR ≤ 0.01 [47].

The correlations between the EBA-O dimensions and respective questionnaires were examined to study the construct validity, considering correlation coefficients (r) ≤ 0.30, as recommended standards.

A two-way multivariate analysis of variance (MANOVA) with the five dimensions of the EBA-O as dependent variables and sex and categorical BMI as independent factors was carried out. Eta-squared (η^2^) was used to measure the effect size of MANOVA, considering values of 0.01, 0.06, and 0.14 as indicating small, medium, and large effects, respectively. The Bonferroni correction was used to correct for multiple comparisons (*p* = 0.05/10 = 0.005).

## 3. Results

After excluding participants who did not meet all inclusion and exclusion criteria, a final sample of 384 participants (154 men, 40.1%; 230 women, 59.9%) was analyzed. Table 1 summarizes the characteristics of this sample.

### 3.1. Confirmatory Factor Analysis

Descriptive statistics for all EBA-O items are shown in Table 2. The confirmatory factor analysis demonstrated excellent fit indices, as follows: CFI = 0.99, TLI = 0.98, RMSEA = 0.03, relative chi-square (χ^2^/*df*) = 2.59, *p* = 0.002 supporting the tested second-order five-factor of the Spanish version of the EBA-O model (Figure 1).

### 3.2. Measurement of Invariance Across Sex

Multiple-group CFA was run to examine the measurement invariance across sex. Appendix A collects the fit indices for the three models and the differences between the pairs of nested models.

First, configural invariance was assessed by estimating both sex groups without equality constraints. The results supported the configural invariance of the EBA-O (M1), as indicated by good fit indices (χ^2^ = 673.00, CFI = 0.98, TLI = 0.98, RMSEA = 0.07, SRMR = 0.08).

Next, metric invariance was tested by constraining factor loadings to be equal across male and female groups. Compared to M1, the metric model (M2) showed a slightly worse fit, but the change in model fit was within acceptable limits. The chi-square difference test was not significant (Δχ^2^ = 18.801, Δ*df* = 17), and the changes in CFI (0.001), TLI (0.003), RMSEA (0.003), and SRMR (−0.001) all fell below the recommended cutoffs, supporting metric invariance.

Finally, scalar invariance was tested by additionally constraining the intercepts across groups (M3). Again, the model fit was slightly worse, but the differences remained within acceptable thresholds. The chi-square difference was not significant (Δχ^2^ = 24.243, Δ*df* = 12), and all changes in fit indices were negligible (ΔCFI = 0.000, ΔTLI = −0.001, ΔRMSEA = 0.004, ΔSRMR = 0.000), supporting scalar invariance across sex.

### 3.3. Measurement of Invariance Across BMI

Measurement invariance across BMI was examined using multiple-group CFA. Participants were grouped into the following two categories based on BMI: overweight (25–29.9) and obese (≥30). The fit indices and model comparisons are summarized in Appendix A.

First, configural invariance (M1) was tested with no equality constraints across the two BMI groups. The model demonstrated acceptable fit (χ^2^ = 707.98, CFI = 0.93, TLI = 0.91, RMSEA = 0.07, SRMR = 0.08), indicating that the factor structure was consistent across groups.

Next, metric invariance (M2) was tested by constraining factor loadings to be equal between the groups. Compared to the configural model, the fit remained stable, with Δχ^2^ = 21.706 (Δ*df* = 17), and minimal changes in fit indices (ΔCFI = 0.001, ΔTLI = 0.000, ΔRMSEA = −0.001, ΔSRMR = −0.001), supporting metric invariance.

Finally, scalar invariance (M3) was tested by additionally constraining item intercepts. Compared to M2, the chi-square difference was not significant (Δχ^2^ = 38.872, Δ*df* = 12), and changes in fit indices (ΔCFI = −0.002, ΔTLI = 0.001, ΔRMSEA = 0.001, ΔSRMR = 0.001) were within recommended thresholds. These results support scalar invariance of the measure across BMI categories.

### 3.4. Internal Consistency

The internal consistency for the EBA-O total score was high (McDonald’s ω = 0.80), indicating excellent reliability of the Spanish version. McDonald’s ω for all factors indicated good reliability, as follows: food addiction (ω = 0.88), night eating (ω = 0.65), binge eating (ω = 0.83), sweet eating (ω = 0.79), and hyperphagia (ω = 0.82). The factors were highly correlated, with the highest correlation observed between factors 1 and 3 and the lowest between factors between 4 and 5 (Table 3).

### 3.5. Concurrent Validity

Correlation analysis (Table 4) revealed significant correlations between the EBA-O subscales and the BES (from r = 0.401 to 0.713), YFAS (from r = 0.342 to 0.730), and EDE-Q total score (from r = 0.285 to 0.699).

### 3.6. Two-Way MANOVA

Significant differences in EBA-O subscales emerged according to sex (F = 3.559, *p* = 0.005; Wilk’s lambda = 0.881; η^2^ = 0.12), BMI category (F = 3.191, *p* < 0.001; Wilk’s lambda = 0.712; η^2^ = 0.11), and their interaction (F = 2.510, *p* = 0.002; Wilk’s lambda = 0.763; η^2^ = 0.09). Sex also had a significant effect on the night eating subscale (F = 12.523, *p* = 0.001; η^2^ = 0.08). Similarly, categorical BMI significantly influenced both food addiction (F = 5.300, *p* = 0.002; η^2^ = 0.11) and night eating (F = 12.685, *p* < 0.001; η^2^ = 0.22) subscales. The means and standard deviations of EBA-O factors and the total score are presented in Table 2.

## 4. Discussion

Obesity is a major public health concern worldwide due to its high prevalence and its contribution to increased morbidity and mortality. Identifying and characterizing eating behaviors is essential for the development of effective and individualized therapeutic interventions. From our understanding, this is the initial attempt to validate an instrument designed to comprehensively assess eating behaviors in individuals with overweight or obesity from a native Spanish-speaking population. The present research focused on the validation and factorial structure of the Spanish version of the Eating Behavior Assessment for Obesity (EBA-O) in a sample of Spanish speaking adults with BMI ≥ 25 kg/m^2^.

Our results demonstrate that the Spanish version of the EBA-O is a valid and reliable instrument comparable to the original Italian version. Confirmatory factor analysis supported a second-order five-factor model, consistent with previous validations. The internal consistency of the instrument was strong, with global McDonald’s ω coefficient = 0.80 ranging from 0.65 to 0.88 for the subscales. Present results are in line with those reported by Segura-García et al. and Mavrandrea et al., affirming the clinical utility of the EBA-O in Spanish-speaking populations [32,33].

The Spanish version of the EBA-O also showed satisfactory convergent validity, as evidenced by positive and significant correlations with other well-established measures of pathological eating behavior, including the BES, the YFAS, and the EDE-Q. These results further support the instrument’s appropriateness for assessing eating behaviors in this population.

The two-way MANOVA brought out significant differences in EBA-O subscale according to sex and BMI categories. Specifically, sex significantly influenced the night eating subscale (F = 12.523, *p* = 0.001, η^2^ = 0.08), with women reporting higher scores than men. Some studies have reported sex differences in night eating patterns [48], whereas others found no consistent differences or suggested that these differences may not be stable over time [49]. Very recently Oteri et al. [50] observed that while the average scores of night eating were significantly higher among males, females exhibited a higher prevalence of night eating (i.e., night eating score ≥ 4). Additionally, the BMI categories significantly influenced both food addiction (F = 5.300, *p* = 0.002, η^2^ = 0.11) and night eating (F = 12.685, *p* < 0.001, η^2^ = 0.22) subscales, with higher BMI categories associated with higher scores on both measures. Oteri et al. [50] found that a higher BMI is associated with greater severity of altered eating behaviors, especially food addiction and binge eating, while hyperphagia is related to slower weight loss during inpatient stay. In our study, the interaction between sex and BMI was also significant (F = 2.510, *p* = 0.002, η^2^ = 0.09), suggesting that these factors may jointly influence eating behaviors. As such interactions effects have not been consistently reported in previous research [32,33], further studies with larger and more diverse samples are warranted to clarify these relationships.

The use of a rigorous back-translation methodology accounting for the relevancy of multiple Spain-specific idioms ensured linguistic and conceptual comprehensibility and accessibility and should be acknowledged as the main strength of the study. The recruitment methodology (i.e., social media platforms) facilitated the widespread diffusion of the survey, boosting accessibility and enabling to participate anonymously and in a non-stigmatizing setting. This is particularly valuable for studies involving individuals with large bodies, for whom stigma and privacy concerns could deter engagement in in-person research environments. On the other hand, the use of convenience sampling via social media might have introduced self-selection, coverage, nonresponse, or volunteer bias. Further, while the sample was sufficient for enabling factor analyses and psychometric validation (statistical adequacy), it may not fully reflect the socio-economic and ethnic heterogeneity of the broader Spanish-speaking population. These two aspects might have limited sample representativeness. Future studies are encouraged to measure invariance across more socio-demographically diverse subgroups to enhance generalizability of the instrument. The absence of a retest assessment limits the evaluation of the temporal stability of the Spanish version of the EBA-O scores. Additionally, future research should evaluate the ability of the Spanish EBA-O to effectively capture and differentiate changes in eating behavior within clinical populations, as demonstrated by previous studies that highlight its efficacy [51,52].

## 5. Conclusions

In conclusion, the Spanish version of the EBA-O is a valid and reliable tool for assessing disordered eating behaviors in Spanish-speaking individuals with overweight or obesity. Its robust psychometric characteristics support its use in both clinical and research contexts. By providing a comprehensive and easy-to-use measure, the EBA-O can assist clinicians in identifying maladaptive eating patterns and tailoring interventions more effectively for this population (Table 5).

## Figures and Tables

**Figure 1 nutrients-17-02344-f001:**
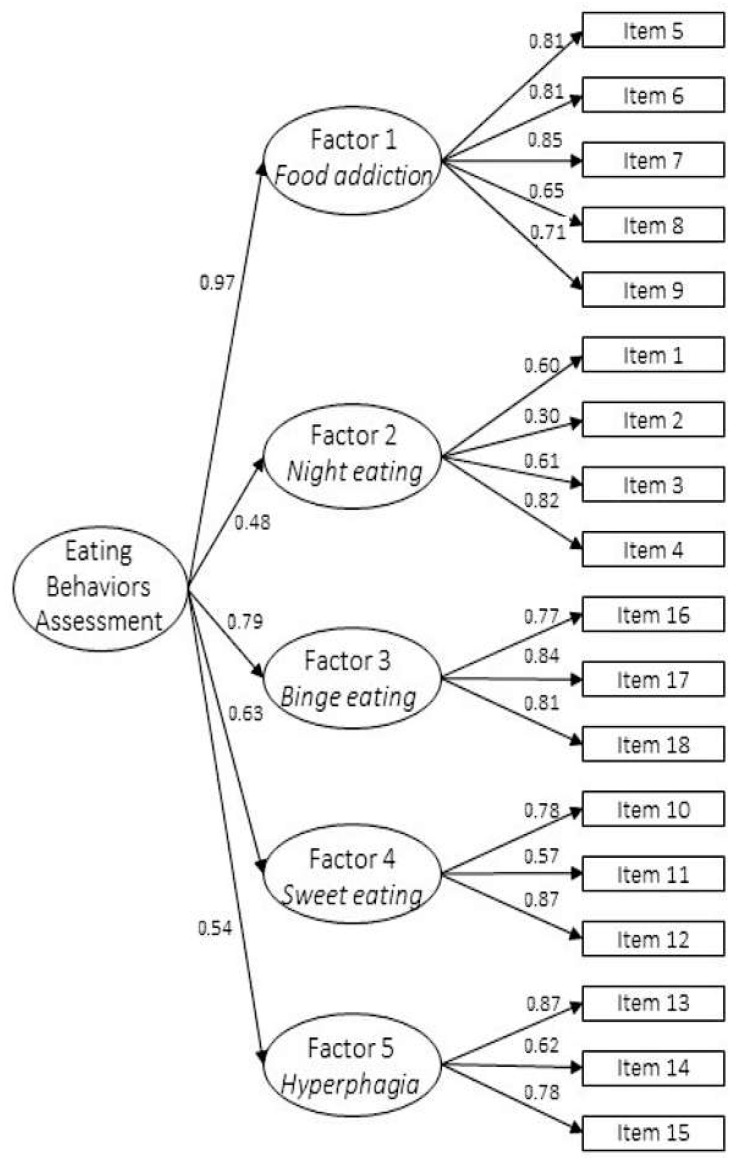
Path diagram of the second-order five-factor model of the EBA-O with reported standardized coefficients of first- and second-order loadings. Adjusted *p* < *0*.001.

**Table 1 nutrients-17-02344-t001:** Socio demographic features of the sample.

		Fr	%
Sex	Female	230	59.9
	Male	154	40.1
Age		37.2	15.8
BMI *		30.8	4.2
BMI category	25–30	277	72.2
	30–35	75	19.4
	35–40	19	4.9
	>40	13	3.5
Education	High school	109	28.4
	University	275	71.6
Occupation	Housewife	5	1.3
	Not working	17	4.4
	Working	197	51.3
	Retired	10	2.6
	Student	155	40.4

* Mean and standard deviation. BMI: body mass index, weight (kg)/[height (m)]^2^; Fr: frequency.

**Table 2 nutrients-17-02344-t002:** Descriptive statistics for each item and subscales of the EBA-O in the current sample (N = 384).

	N	Min	Max	Mean	SD	Skewness	Kurtosis
Item 1	384	0	7	0.49	1.167	0.803	2.500
Item 2	384	0	7	0.43	1.433	0.982	2.936
Item 3	384	0	7	0.93	1.645	0.852	2.226
Item 4	384	0	7	0.55	1.294	0.749	2.632
Item 5	384	0	7	1.87	1.935	0.818	−0.362
Item 6	384	0	7	1.25	1.734	1.361	0.933
Item 7	384	0	7	1.38	1.883	1.335	0.865
Item 8	384	0	7	1.51	2.147	1.391	0.736
Item 9	384	0	7	0.73	1.574	1.256	2.498
Item 10	384	0	7	2.83	2.202	0.465	−0.908
Item 11	384	0	7	3.48	2.123	−0.059	−0.949
Item 12	384	0	7	1.92	2.149	0.860	−0.409
Item 13	384	0	7	1.64	2.273	1.170	0.019
Item 14	384	0	7	1.74	1.922	0.867	−0.343
Item 15	384	0	7	2.30	2.044	0.518	−0.760
Item 16	384	0	7	1.49	2.032	1.273	0.494
Item 17	384	0	7	0.90	1.804	0.993	1.546
Item 18	384	0	7	0.94	2.033	0.753	2.589
Food addiction	384	0	7	1.35	1.514	1.324	1.299
Night eating	384	0	6	0.60	0.991	2.318	2.687
Binge eating	384	0	7	1.11	1.724	1.767	2.285
Sweet eating	384	0	7	2.74	1.807	0.464	−0.449
Hyperphagia	384	0	7	1.89	1.760	0.896	−0.088
Total score	384	0	6.6	1.54	1.12	1.238	1.711

**Table 3 nutrients-17-02344-t003:** Correlations between factors of the SP-EBA-O.

	Factor 1	Factor 2	Factor 3	Factor 4	Factor 5
Factor 1 Food addiction	—				
Factor 2 Night eating	0.367 ***	—			
Factor 3 Binge eating	0.638 ***	0.246 ***	—		
Factor 4 Sweet eating	0.573 ***	0.256 ***	0.338 ***	—	
Factor 5 Hyperphagia	0.402 ***	0.189 ***	0.547 ***	0.188 ***	—

*** *p*< 0.001.

**Table 4 nutrients-17-02344-t004:** Results of convergent validity.

	Factor 1 Food Addiction	Factor 2 Night Eating	Factor 3 Binge Eating	Factor 4 Sweet Eating	Factor 5 Hyperphagia
BES	0.713 **	0.401 **	0.771 **	0.496 **	0.464 **
YFAS 2.0 Symptom count	0.730 **	0.342 **	0.701 **	0.447 **	0.459 **
EDE-Q Total score	0.659 **	0.285 **	0.699 **	0.405 **	0.356 **

BES: Binge Eating Scale, YFAS 2.0: Yale Food Addiction Scale 2.0. ** *p* < 0.001.

**Table 5 nutrients-17-02344-t005:** Features of the EBA-O.

Reliable: Specifically designed for individuals with obesity.
**Comprehensive**: Allows easy assessment of common eating behaviors associated with obesity.
**User-friendly**: Simple to interpret, even for clinicians without specialization in eating disorders.
**Efficient**: Assesses five key eating behaviors using only a few targeted questions.
**Quick Administration**: Fast and easy to administer in clinical settings.
**Quick Scoring**: Rapid and straightforward to score.
**Educational**: Valuable for patient psychoeducation.
**Risk Identification**: Helps identify individuals at risk for eating disorders.
**Support Detection**: Identifies those who may benefit from additional psychological support.
**Progress Monitoring**: Useful for tracking changes in eating behaviors during treatment.

## Data Availability

Data are available from the corresponding author upon request. The data are not publicly available due to.

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
