# Peer review of "Bridging Gaps in Obesity Assessment: Spanish Validation of the Eating Behaviors Assessment for Obesity (EBA-O)"

_nutrients, 2025, doi:10.3390/nu17142344_

Round 1

Reviewer 1 Report (Previous Reviewer 2)

Comments and Suggestions for Authors

Dear Authors,

I would like to commend the authors for the substantial revisions made to the manuscript. The improvements in structure and clarity are evident, and the effort to address prior concerns is appreciated.

Sorry, if I missed that, but the sample size justification is still absent. Also, while I understand the authors’ rationale for proceeding directly to CFA based on the original theoretical model of the EBA-O, I would encourage the authors to provide a methodological reference that supports the omission of EFA in the context of cross-cultural scale adaptation. Standard psychometric practice typically recommends EFA when an instrument is translated and adapted to a new cultural or linguistic context.

Regarding Figure 2, I suggest considering a replacement with a descriptive table that includes more detailed information about the subscale scores. Given the distribution characteristics, presenting not only means and standard deviations but also additional metrics (e.g., medians, ranges, Skewness, Kurtosis) could be highly informative for researchers who may use this scale in future studies. While I am not certain whether it is allowed, I have attached an anonymous sample table for illustrative purposes (if not allowed, the editors will remove it). That said, this is a minor point, and the decision ultimately rests with the authors.

All the best.

Author Response

I would like to commend the authors for the substantial revisions made to the manuscript. The improvements in structure and clarity are evident, and the effort to address prior concerns is appreciated.

Answer: Thank you again for the time you have dedicated to our manuscript. We do believe that the comments and suggestions you have given us have been really helpful. We hope to have addressed all your suggestions.

Sorry, if I missed that, but the sample size justification is still absent.

Answer: We have added the justification for the sample size according to G-Power “According to MacCallum et al. (1996) based on power analysis using RMSEA-based criteria a sample size of 380 was deemed sufficient, assuming moderate to high factor loadings (≥ 0.5) and model degrees of freedom ≈130. The power to detect close versus poor fit (RMSEA 0.05 vs. 0.08) with α = 0.05 was estimated to be >0.85, indicating that the sample size was adequate to detect model misfit and ensure stable parameter estimates (Wolf et al., 2013).

References:

  • MacCallum, R. C., Browne, M. W., & Sugawara, H. M. (1996). Power analysis and determination of sample size for covariance structure modeling. Psychological Methods, 1(2), 130–149.
  • Wolf, E. J., Harrington, K. M., Clark, S. L., & Miller, M. W. (2013). Sample size requirements for structural equation models: An evaluation of power, bias, and solution propriety. Educational and Psychological Measurement, 73(6), 913–934.

Also, while I understand the authors’ rationale for proceeding directly to CFA based on the original theoretical model of the EBA-O, I would encourage the authors to provide a methodological reference that supports the omission of EFA in the context of cross-cultural scale adaptation. Standard psychometric practice typically recommends EFA when an instrument is translated and adapted to a new cultural or linguistic context.

Answer: We appreciate the reviewer’s thoughtful comment regarding the use of confirmatory factor analysis (CFA) without preceding exploratory factor analysis (EFA) in the context of cross-cultural adaptation.

Our decision to proceed directly to CFA was guided by the fact that the Eating Behaviors Assessment for Obesity (EBA-O) is a strongly theory-driven instrument with a well-established factor structure grounded in empirical research. As such, our adaptation aimed to test whether the original theoretical model could be confirmed in the new cultural context, rather than to explore an alternative structure.

This approach is consistent with the recommendations of Brown (2006 and 2015), who emphasizes that CFA is appropriate when a strong a priori theoretical model exists. According to Brown, "unlike EFA, CFA requires a strong empirical or conceptual foundation to guide the specification and evaluation of the factor model" (p. 12). Moreover, CFA is particularly well-suited for testing the applicability of a known model in new populations, including cross-cultural contexts, provided that the theoretical basis of the model remains relevant.

This is the reference: “Brown, T. A. (2015). Confirmatory Factor Analysis for Applied Research (2nd ed.). The Guilford Press”.

Nevertheless, here we show the EFA to further test if the Spanish version confirmed the Italian results. The results confirm the original 5 factors model with the same items as follows:

Factor Loadings

FACTOR 1

FACTOR 2

FACTOR 3

FACTOR 4

FACTOR 5

Uniqueness

Item 6

0.963

0.159

Item 5

0.730

0.312

Item 7

0.660

0.303

Item 9

0.448

0.531

Item 8

0.398

0.536

Item 18

0.926

0.137

Item 17

0.825

0.208

Item 16

0.428

0.447

Item 15

0.857

0.242

Item 14

0.726

0.488

Item 13

0.595

0.446

Item 11

0.802

0.401

Item 10

0.778

0.307

Item 12

0.483

0.454

Item 4

0.802

0.357

Item 3

0.604

0.615

Item 1

0.596

0.616

Item 2

0.421

0.830

Factor Characteristics

Unrotated solution

Rotated solution

Eigenvalues

SumSq. Loadings

Proportion var.

Cumulative

SumSq. Loadings

Proportion var.

Cumulative

Factor 1

6.594

6.242

0.347

0.347

2.770

0.154

0.154

Factor 2

2.036

1.661

0.092

0.439

2.358

0.131

0.285

Factor 3

1.691

1.172

0.065

0.504

1.981

0.110

0.395

Factor 4

1.281

0.948

0.053

0.557

1.740

0.097

0.492

Factor 5

0.968

0.565

0.031

0.588

1.738

0.097

0.588

Regarding Figure 2, I suggest considering a replacement with a descriptive table that includes more detailed information about the subscale scores. Given the distribution characteristics, presenting not only means and standard deviations but also additional metrics (e.g., medians, ranges, Skewness, Kurtosis) could be highly informative for researchers who may use this scale in future studies. While I am not certain whether it is allowed, I have attached an anonymous sample table for illustrative purposes (if not allowed, the editors will remove it). That said, this is a minor point, and the decision ultimately rests with the authors.

Answer: We thank the reviewer for this suggestion. In the previous draft, we had included this table within the Supplementary material. Now, we have replaced Figure 2 with this descriptive table that includes detailed information for each item, subscale, and the total score of the EBA-O. The table reports means, standard deviations, ranges, skewness, and kurtosis, as suggested. We believe this provides a more comprehensive overview of the distributional characteristics and will be valuable to future researchers using the scale.

Table 2 .  Descriptive statistics for each Item and subscales of the EBA-O in the current sample (N = 384)

N

Min

Max

Mean

SD

Skewness

Kurtosis

Item 1

384

0

7

0.49

1.167

0.803

2.500

Item 2

384

0

7

0.43

1.433

0.982

2.936

Item 3

384

0

7

0.93

1.645

0.852

2.226

Item 4

384

0

7

0.55

1.294

0.749

2.632

Item 5

384

0

7

1.87

1.935

0.818

-0.362

Item 6

384

0

7

1.25

1.734

1.361

0.933

Item 7

384

0

7

1.38

1.883

1.335

0.865

Item 8

384

0

7

1.51

2.147

1.391

0.736

Item 9

384

0

7

0.73

1.574

1.256

2.498

Item 10

384

0

7

2.83

2.202

0.465

-0.908

Item 11

384

0

7

3.48

2.123

-0.059

-0.949

Item 12

384

0

7

1.92

2.149

0.860

-0.409

Item 13

384

0

7

1.64

2.273

1.170

0.019

Item 14

384

0

7

1.74

1.922

0.867

-0.343

Item 15

384

0

7

2.30

2.044

0.518

-0.760

Item 16

384

0

7

1.49

2.032

1.273

0.494

Item 17

384

0

7

0.90

1.804

0.993

1.546

Item 18

384

0

7

0.94

2.033

0.753

2.589

Food addiction

384

0

7

1.35

1.514

1.324

1.299

Night eating

384

0

6

0.60

0.991

2.318

2.687

Binge eating

384

0

7

1.11

1.724

1.767

2.285

Sweet eating

384

0

7

2.74

1.807

0.464

-0.449

Hyperphagia

384

0

7

1.89

1.760

0.896

-0.088

Total score

384

0

6.6

1.54

1.12

1.238

1.711

Reviewer 2 Report (Previous Reviewer 1)

Comments and Suggestions for Authors

Dear authors, 

I have read your article entitled "Bridging Gaps in Obesity Assessment: Spanish Validation of the Eating Behaviors Assessment for Obesity (EBA-O)." The validation of this type of questionnaire is useful, taking into account that obesity cases are rising. I have attached my suggestions regarding your article below.

1. Please attach the ethical approval paper translated into English.
2. Please attach the author's consent for translation and adaptation.
3. The number of women and men is quite different. There are also significant differences in work and education. Do you consider that the lack of homogeneity between the target groups can yield valid results?
4. How would you like to use this questionnaire in the future? Describe in a subchapter its practical usefulness.
5. What are the limitations and strengths of this study?
6. Please include a graph with the number of participants recruited from each platform used, together with their general data.

Author Response

I have read your article entitled "Bridging Gaps in Obesity Assessment: Spanish Validation of the Eating Behaviors Assessment for Obesity (EBA-O)." The validation of this type of questionnaire is useful, taking into account that obesity cases are rising. I have attached my suggestions regarding your article below.

Answer: Thank you for the time you have dedicated to our manuscript and the comments and suggestions you have given us. We have tried to address all your comments and suggestions.

  1. Please attach the ethical approval paper translated into English.

Answer: According to this request we are providing at the end of the answers the Spanish and English versions of:

  • ETHICAL APPROVAL from the Ethics Committee of University of Cordoba
  • INFORMED CONSENT – INFORMATION FOR THE PARTICIPANT
  • INFORMED CONSENT – WRITTEN CONSENT OF THE PARTICIPANT
  1. Please attach the author's consent for translation and adaptation.

Answer: Dear Reviewer, we understand your request and appreciate the attention to proper procedures. However, with all due respect, as the first author of the original Italian version of the EBA-O, and at the same time, the corresponding author of the current manuscript, I find it a bit paradoxical to seek authorization from myself to translate and adapt a tool I personally developed, into my own native language.

  1. The number of women and men is quite different. There are also significant differences in work and education. Do you consider that the lack of homogeneity between the target groups can yield valid results?

Answer: Thank you for this interesting question. While there are certainly differences in the distribution of gender, education, and employment status among participants, the results of the Spanish version of the EBA-O can still be considered valid. Specifically, metric invariance between genders was confirmed (Supplementary tables), indicating that men and women interpreted and responded to the scale items ina comparable manner. This suggests that the underlying constructs measured by the EBA-O function equally across genders, supporting the validity of the results across gender. Regarding educational differences, although there is variation in educational level, education is not expected to systematically influence the fundamental eating behaviors of people with obesity. The EBA-O is designed to capture the psychological and behavioral aspects of eating characteristic of obesity, which are generally considered relatively independent of formal education level. Therefore, although demographic heterogeneity exists, this does not compromise the validity of the scale's results in this context.

  1. How would you like to use this questionnaire in the future? Describe in a subchapter its practical usefulness.

Answer: The EBA-O is already used in other languages both for clinical and research purposes, so authors’ opinion is that the Spanish version will be helpful for clinicians not only to characterize patients’ eating behavior in order to taylor treatment but also to assess the evolution of eating behaviors under treatment. As suggested by one reviewer we had included a last table (Table 5) with the main pros of this tool.

  1. What are the limitations and strengths of this study?

Answer: In the last paragraph of the discussion, before the conclusions, we had included the pros and cons of our study “The use of a rigorous back-translation methodology accounting for the relevancy of multiple Spain-specific idioms ensured linguistic and conceptual comprehensibility and accessibility and should be acknowledged as the main strenght of the study. The recruitment methodology (i.e., social media platforms) facilitated widespread diffusion of the survey, boosting accessibility and enabling to participare anonymously and in a non-stigmatizing setting. This is particularly valuable for studies involving individuals suffering from overweight or obesity, for whom stigma and privacy concerns could deter engagement in in-person research environments. On the other hand, the use of convenience sampling via social media might have introduced self-selection, coverage, nonresponse or volunteer bias. Further, while the sample was sufficient for enabling factor analyses and psychometric validation (statistical adequacy), it may not fully reflect the socio-economic and ethnic heterogeneity of the broader Spanish-speaking population. These two aspects might have limited sample representativeness. Future studies are encouraged to measure invariance across more socio-demographically diverse subgroups to enhance generalizability of the instrument. The absence of a retest assessment limits the evaluation of the temporal stability of the Spanish version of the EBA-O scores. Additionally, future research should evaluate the ability of the Spanish EBA-O to effectively capture and differentiate changes in eating behavior within clinical populations, as demonstrated by previous studies that highlight its efficacy [36,37].

  1. Please include a graph with the number of participants recruited from each platform used, together with their general data.

Answer: Thank you very much for this question. Unfortunately, it is not possible for us to provide an answer as all participants answer to the same link even if the advertisements were done in several platforms on the internet. Nevertheless, your suggestion will be considered in our future studies with the same procedure.

Round 2

Reviewer 2 Report (Previous Reviewer 1)

Comments and Suggestions for Authors

The article can be published. Congratulations!

This manuscript is a resubmission of an earlier submission. The following is a list of the peer review reports and author responses from that submission.

Round 1

Reviewer 1 Report

Comments and Suggestions for Authors

Dear authors,

I read your article; it is well-written and structured, but a few adjustments and additions are necessary for publication.

1. Considering that this is an English article, I believe that the authors should also provide the English version of Table 1 if possible. 
2. To improve the quality of the article, the authors should add a graphical abstract. 
3. The introduction is well-written and quite lengthy. 
4. The materials and methods chapter is clear and well-structured. 
5. Do you consider that the sample of 384 participants was representative of the entire obese Spanish population, taking into account socio-economic and ethnic diversity? Justify. 
6. It would be useful for the authors to add a table or even in the conclusion chapter, 10 specific arguments that support the validity of this tool. 

Author Response

Dear authors,

I read your article; it is well-written and structured, but a few adjustments and additions are necessary for publication.

Answer: The authors thank the Reviewer for the time spent on the manuscript and for the valuable feedback provided. Hereby addressed the comments point by point.

Considering that this is an English article, I believe that the authors should also provide the English version of Table 1 if possible. 

Answer: The authors would like to respectfully clarify that the Italian version of the EBA-O served as the basis for the forward/backward translation process and the development of the Spanish version. However, the original EBA-O validation article in Italian also contains the items in English.

To improve the quality of the article, the authors should add a graphical abstract. 

Answer: Thank you very much for this suggestion, we have included a graphical abstract in this new version.

The introduction is well-written and quite lengthy. 

Answer: The authors addressed the other Reviewer’s concerns about the need for additional contents in the introduction, while carefully saving words and maintaining a focused approach. We trust the Reviewer will find the revised paragraph more informative, still well-written, and appropriately concise.

The materials and methods chapter is clear and well-structured.

Answer: This paragraph has been revised to incorporate both minor and substantial suggestions received from the Editor. Additional details regarding the procedures, informed consent and sample size have been included in the Participants section. Furthermore, the Statistical Analysis section has been implemented to include a multiple-group confirmatory factor analysis, aimed at exploring the invariance of the EBA-O across sex and BMI. All changes are marked in red for the Reviewer's convenience.

Do you consider that the sample of 384 participants was representative of the entire obese Spanish population, taking into account socio-economic and ethnic diversity? Justify. 

Answer: The authors thank the reviewer for raising this point and the chance to justify the sample's relevance according to the study’s aims. The sample was not drawn to be nationally representative beyond the key demographic variables (e.g., age, sex, BMI). While a sample of three hundreds is generally sufficient to conduct a robust psychometric validation (statistical adequacy) (N=384) (MacCallum et al., 1999; Hair et al., 2010), the authors acknowledge that it might not have fully captured the heterogeneity the Spanish-speaking population express when it comes to socio-economic and ethnic diversity (population representativeness). Aware of the relevancy of multiple Spain-specific idioms, neutral and standard Spanish during the translation process has been applied. To boost its applicability in wider Spanish-speaking contexts, the preliminary translation was further filtered by a group of 15 volunteers from various Spanish-speaking countries (i.e., Argentina, Chile, Colombia, Mexico and Spain). For the purpose of initial validation of the EBA-O for people self-identifying Spanish as its primary language, the authors believe this sample is diverse enough to reflect variation in psychometric properties. We have now added explicitly mentioned the limitation and discussed the need for future research to assess measurement invariance across more socio-demographically diverse Spanish-speaking subpopulations.

It would be useful for the authors to add a table or even in the conclusion chapter, 10 specific arguments that support the validity of this tool. 

Answer: The authors thank the reviewer for this suggestion. We have provided a table with ten specific arguments that support the validity of this tool

Table 4. Features of the EBA-O

· Reliable: Specifically designed for individuals with obesity.

· Comprehensive: Allows easy assessment of common eating behaviors associated with obesity.

· User-friendly: Simple to interpret, even for clinicians without specialization in eating disorders.

· Efficient: Assesses five key eating behaviors using only a few targeted questions.

· Quick Administration: Fast and easy to administer in clinical settings.

· Quick Scoring: Rapid and straightforward to score.

· Educational: Valuable for patient psychoeducation.

· Risk Identification: Helps identify individuals at risk for eating disorders.

· Support Detection: Identifies those who may benefit from additional psychological support.

· Progress Monitoring: Useful for tracking changes in eating behaviors during treatment.

Reviewer 2 Report

Comments and Suggestions for Authors

This manuscript attempts to validate the Spanish version of the Eating Behaviors Assessment for Obesity (EBA-O) in a sample of adults with overweight or obesity. While the topic is clinically relevant and potentially impactful, the manuscript suffers from several critical methodological and conceptual flaws that substantially undermine its validity and scientific contribution.

Major Concerns:

  1. The Introduction does not adequately justify the need for a Spanish version of the EBA-O. It fails to mention existing instruments available in Spanish that assess similar constructs, nor does it explain why a new validation is essential. The importance of validating this specific instrument (versus others) for Spanish-speaking populations is not convincingly argued. No background is provided on the psychometric performance of the EBA-O in other languages (e.g., Italian, Greek), which would provide context and rationale for cross-cultural adaptation.

  2. The authors proceed directly to Confirmatory Factor Analysis (CFA) without performing Exploratory Factor Analysis (EFA), which is particularly concerning given the adaptation to a new language and cultural context. No test of measurement invariance is conducted before conducting group comparisons (e.g., across sex or BMI groups). This omission is a significant methodological flaw, as such analyses assume metric and scalar invariance across groups. The study makes cross-group comparisons (via MANOVA) based on potentially non-invariant factors, which undermines the validity of these comparisons.

  3. There is no justification or power analysis provided for the sample size used (N = 384). Given the use of CFA and subgroup analyses, a rationale is essential.

  4. The description of study ethics procedures and informed consent is insufficient in the Methods section. Although ethical approval is briefly mentioned later in the manuscript, the process by which informed consent was obtained from participants should be clearly stated within the methodological framework. Moreover, the data collection strategy raises concerns about sample quality and representativeness. Recruitment via open sharing of the survey link on social media platforms introduces a significant risk of sampling bias, as such approaches often lead to self-selected, non-representative samples. Additionally, this method lacks controls to verify participant eligibility or to prevent duplicate or inattentive responses, potentially compromising the validity and reliability of the data collected.
  5. While the BES, YFAS, and EDE-Q are used to establish convergent validity, the manuscript fails to confirm that validated Spanish versions of these instruments were employed or to reference those validations. BMI is not included in the study measures description.

  6. The limitations section is inadequate, failing to acknowledge the lack of invariance testing and the absence of EFA.

Minor Issues:

  1. Figure 1 (subscale means) adds little interpretative value. Descriptive statistics should be reported in a table with means, SDs, Skewness, Kurtosis, min., max. values and ranges. Table 1 includes the full Spanish translation of the scale within the main text. This would be more appropriate as an Appendix. Table 2 omits the unit of measurement for BMI (e.g., kg/m²), a basic reporting standard.

  2. There are multiple typographical and formatting issues throughout the manuscript, which detract from its readability.

Comments on the Quality of English Language

English requires extensive editing.

Author Response

This manuscript attempts to validate the Spanish version of the Eating Behaviors Assessment for Obesity (EBA-O) in a sample of adults with overweight or obesity. While the topic is clinically relevant and potentially impactful, the manuscript suffers from several critical methodological and conceptual flaws that substantially undermine its validity and scientific contribution.

Answer: The authors sincerely thank the Reviewer for the time dedicated on evaluating the manuscript and for the valuable and constructive feedback provided. Authors made a range of revisions—some minor, others more substantial— to meet the Reviewer’s suggestions. Authors trust the revised manuscript has significantly benefited from the review process and now better aligns with the conceptual and methodological prerequisites expected for a validation study. Below provided a point-by-point response to each comment.

Major Concerns:

The Introduction does not adequately justify the need for a Spanish version of the EBA-O. It fails to mention existing instruments available in Spanish that assess similar constructs, nor does it explain why a new validation is essential. The importance of validating this specific instrument (versus others) for Spanish-speaking populations is not convincingly argued. No background is provided on the psychometric performance of the EBA-O in other languages (e.g., Italian, Greek), which would provide context and rationale for cross-cultural adaptation.

Answer: Thank you for this comment. In the introduction we wrote: “…However, assessing these varied eating behaviors can be challenging in clinical settings. Currently available tools often require the use of multiple scales, each designed to evaluate a single behavior, and typically tailored to patients with eating disorders rather than those with obesity. This fragmented approach is time-consuming and may be difficult to interpret, especially for clinicians lacking specialized training in eating disorders. Therefore, there is a clear need for a comprehensive, reliable, and easy-to-use instrument to streamline the assessment of disordered eating behaviors in this population…”, so the authors respectfully consider that they have sufficiently justified the need for an ad hoc scale for people with obesity. Anyway, the authors agree with the reviewer that a further explanation that justifies the Spanish validation could be useful, so this paragraph was included in the new draft: “To the authors' knowledge, no specifically designed for people with obesity and validated in Spanish instrument that simultaneously assesses the five eating disorders examined by the EBA-O (i.e., night eating, food addiction, sweet eating, hyperphagia, binge eating) exists.”  To further justify the reason of this validation, and according to the request from Reviewer 1, we have provided a final Table 4 that reassumes the motivations and utility of the validated EBA-O.

Authors have tried to improve the lacking background on the psychometric performance of the EBA-O in Italian and Greek so that to provide context and rationale for cross-cultural adaptation: “Both, Italian and Greek versions, coincide with an 18-item second-order five-factor model, and have proved the EBA-O to have good internal consistency (McDonald’s w ranging from .80 to .92), good convergent validity with other measures of eating psychopathology, and good discriminant validity.

Answer: The authors proceed directly to Confirmatory Factor Analysis (CFA) without performing Exploratory Factor Analysis (EFA), which is particularly concerning given the adaptation to a new language and cultural context. No test of measurement invariance is conducted before conducting group comparisons (e.g., across sex or BMI groups). This omission is a significant methodological flaw, as such analyses assume metric and scalar invariance across groups. The study makes cross-group comparisons (via MANOVA) based on potentially non-invariant factors, which undermines the validity of these comparisons.

Answer: The Authors thank the reviewer for the valuable observation. Regarding the use of Exploratory Factor Analysis (EFA), we would like to clarify that EFA is not strictly required when validating an existing instrument which is grounded in a well-established theoretical model. This is the case for the EBA-O. The aim of the present study is to test the hypothesized factor structure of the EBA-O directly through Confirmatory Factor Analysis (CFA), in line with standard practice for the validation of cross-cultural adaptations of psychometric tools.

Nevertheless, the authors fully agree with the reviewer on the importance of testing measurement invariance prior to conducting group comparisons. In response to this valuable suggestion, we have conducted measurement invariance analyses across sex and categorical BMI groups (normal weight vs. overweight/obesity), examining configural, metric, and scalar invariance. The results of these analyses have been added to the revised manuscript. We sincerely appreciate the reviewer’s feedback; the authors trust this further analysis has enhanced the methodological rigor and the overall statistical adequacy of the study.

There is no justification or power analysis provided for the sample size used (N = 384). Given the use of CFA and subgroup analyses, a rationale is essential.

Answer: Thank the Reviewer for the chance to clarify this point. The authors referenced prior literature suggesting that a sample of 300 is generally considered sufficient for a validation study to enable a robust factor analysis (MacCallum et al., 1999; Hair et al., 2010). While consistent with the statistical adequacy requirement, the authors acknowledge the sample might not have fully captured the heterogeneity the Spanish-speaking population express when it comes to socio-economic and ethnic diversity (population representativeness). We have now addressed this limitation highlighting the need for future research to assess measurement invariance across more socio-demographically diverse subgroups.

The description of study ethics procedures and informed consent is insufficient in the Methods section. Although ethical approval is briefly mentioned later in the manuscript, the process by which informed consent was obtained from participants should be clearly stated within the methodological framework.

Moreover, the data collection strategy raises concerns about sample quality and representativeness. Recruitment via open sharing of the survey link on social media platforms introduces a significant risk of sampling bias, as such approaches often lead to self-selected, non-representative samples.

This method lacks controls to verify participant eligibility or to prevent duplicate or inattentive responses, potentially compromising the validity and reliability of the data collected.

Answer: The participants and procedures section has been implemented and detailed according to Reviewer’s concerns about methodology and informed consent.

The authors acknowledge that recruitment via social media might have introduced a potential for self-selection and sampling bias. However, in the context of psychometric validations, the primary aim is to assess the internal structure, reliability, and construct validity of the instrument over the representativeness of the sample.  Moreover, although accounting for self-selection, coverage, nonresponse and volunteer bias, the authors trust this approach also offered substantial benefits. Specifically, the open sharing via facilitated a widespread diffusion of the survey, boosting accessibility and enabling participation in an anonymous and non-stigmatizing setting. This is particularly valuable for studies involving individuals suffering from overweight or obesity, for whom stigma and privacy concerns could deter engagement in in-person research environments.

We have now added a paragraph to the Discussion section addressing both the strengths and limitations of this approach.

Lastly, eligibility was double checked. Participants were firstly informed about criteria for participation at the beginning of the online survey (e.g., being 18 years or older, fluent in Spanish, and having a self-reported BMI ≥ 25 kg/m²). Then, a thorough post-hoc analysis of participants’ responses was run to further control for eligibility. This approach ensured that the final sample conformed to the methodological standards required for psychometric validation, while maintaining accessibility and ethical transparency. This procedure is now detailed in the participants and procedures section.

While the BES, YFAS, and EDE-Q are used to establish convergent validity, the manuscript fails to confirm that validated Spanish versions of these instruments were employed or to reference those validations. BMI is not included in the study measures description.

Answer: The authors acknowledge they might have underestimated the importance to explicitly state that all the instruments used for the convergent validity analysis were Spanish-language versions with published validation data. However, the references in the manuscript were correctly cited and redirecting to the Spanish versions of the questionnaires (BES: Escrivá-Martínez et al. (2019) [22]; YFAS 2.0: Granero et al. (2018) [23]; EDE-Q: Villarroel et al. (2011) [24]). To improve clarity, the relevant sentence has been rephrased for more clarity (“To assess the convergent validity of the Spanish version of the EBA-O, the following psychometric questionnaires with established validation studies conducted in Spanish-speaking populations were administered”). The Authors have also consistently rephrased the participants and procedures section; the BMI measure is now explicitly stated for clarity and completeness: “the survey collected demographics (i.e., age, sex) and socio-demographic characteristics (i.e., education level, employment status). It also gathered information on general medical history (including current or past mental health disorders and active treatments), and participants were asked to self-report their current weight and height for body mass index calculation (BMI; weight (kg) / [height (m)] ²)”.

The limitations section is inadequate, failing to acknowledge the lack of invariance testing and the absence of EFA.

Answer: The invariance testing has been conducted, included in the revised manuscript and no longer constitute a limitation to address, thanks to the suggestion of the Reviewer.

Regarding the absence of EFA, the authors would respectfully argue this item does not constitute a limitation of the study. As clarified in the previous responses, this analysis was not run because the EBA-O already has an established theoretical model. In line with standard practice for the validation of cross-cultural adaptations of psychometric tools, the original factor structure was tested through Confirmatory Factor Analysis (CFA), and the analysis is now more adequate including the invariance testing.

The limitations section, however, has been considerably implemented from its first version and is now more detailed in accordance with the Reviewer’s feedback on representativeness and statistical adequacy.

Minor Issues:

Figure 1 (subscale means) adds little interpretative value.

Answer: The authors would need some clarification about this comment.

Descriptive statistics should be reported in a table with means, SDs, Skewness, Kurtosis, min., max. values and ranges.

Answer: The authors thank the reviewer for this helpful suggestion. In response, a comprehensive table reporting the descriptive statistics for all EBA-O items (means, standard deviations, skewness, kurtosis, minimum and maximum values) has been added to the revised manuscript (Table S2, supplemental material).

Table 1 includes the full Spanish translation of the scale within the main text. This would be more appropriate as an Appendix.

Answer: The authors thank for this suggestion. The Spanish version of the EBA-O is now embedded within the supplemental material (Table S1).

Table 2 omits the unit of measurement for BMI (e.g., kg/m²), a basic reporting standard.

Answer: BMI measurement is now included in the procedures section and Table 1 (ex-table 2).

There are multiple typographical and formatting issues throughout the manuscript, which detract from its readability.

Answer: The authors thank the reviewer for this comment. An extensive review of English language has ben done by a professional proof reader.

Comments on the Quality of English Language

English requires extensive editing.

Answer: The text has been extensively revised by an English proof-reader.